# Primary Metabolic Response of *Aristolochia contorta* to Simulated Specialist Herbivory under Elevated CO_2_ Conditions

**DOI:** 10.3390/plants13111456

**Published:** 2024-05-24

**Authors:** Hyeon Jin Jeong, Bo Eun Nam, Se Jong Jeong, Gisuk Lee, Sang-Gyu Kim, Jae Geun Kim

**Affiliations:** 1Department of Biology Education, Seoul National University, Seoul 08826, Republic of Korea; hjjeong123@korea.kr (H.J.J.);; 2Division of Forest Biodiversity, Korea National Arboretum, Pocheon 11187, Republic of Korea; 3Research Institute of Basic Science, Seoul National University, Seoul 08826, Republic of Korea; 4Seoul National University Elementary School, Seoul 03087, Republic of Korea; 5Department of Biological Sciences, Korea Advanced Institute for Science and Technology (KAIST), Daejeon 34141, Republic of Korea; 6Center for Education Research, Seoul National University, Seoul 08826, Republic of Korea

**Keywords:** climate change, defense response, folivorous herbivory, insect herbivory, plant growth, primary metabolite, *Sericinus montela*

## Abstract

This study explores how elevated carbon dioxide (CO_2_) levels affects the growth and defense mechanisms of plants. We focused on *Aristolochia contorta* Bunge (Aristolochiaceae), a wild plant that exhibits growth reduction under elevated CO_2_ in the previous study. The plant has *Sericinus montela* Gray (Papilionidae) as a specialist herbivore. By analyzing primary metabolites, understanding both the growth and defense response of plants to herbivory under elevated CO_2_ conditions is possible. The experiment was conducted across four groups, combining two CO_2_ concentration conditions (ambient CO_2_ and elevated CO_2_) with two herbivory conditions (herbivory treated and untreated). Although many plants exhibit increased growth under elevated CO_2_ levels, *A. contorta* exhibited reduced growth with lower height, dry weight, and total leaf area. Under herbivory, *A. contorta* triggered both localized and systemic responses. More primary metabolites exhibited significant differences due to herbivory treatment in systemic tissue than local leaves that herbivory was directly treated. Herbivory under elevated CO_2_ level triggered more significant responses in primary metabolites (17 metabolites) than herbivory under ambient CO_2_ conditions (five metabolites). Several defense-related metabolites exhibited higher concentrations in the roots and lower concentrations in the leaves in response to the herbivory treatment in the elevated CO_2_ group. This suggests a potential intensification of defensive responses in the underground parts of the plant under elevated CO_2_ levels. Our findings underscore the importance of considering both abiotic and biotic factors in understanding plant responses to environmental changes. The adaptive strategies of *A. contorta* suggest a complex response mechanism to elevated CO_2_ and herbivory pressures.

## 1. Introduction

Climate change exerts a significant influence on the ecological and physiological characteristics of plants. One of the primary factors causing climate change is the increasing concentration of carbon dioxide (CO_2_). As a greenhouse gas, CO_2_ traps heat in the atmosphere, which is called greenhouse effect. Greenhouse effect is essential for life as it keeps the earth warm enough to be habitable [1]. However, human activities, such as burning of fossil fuels, industrial process, and deforestation have dramatically increased the concentrations of CO_2_ in the atmosphere. It results in shifts in temperature, which impact the timing and rate of plant growth [2,3]. These environmental changes disrupt metabolic processes, altering enzyme activities and biochemical pathways within plants. Furthermore, the influences of climate change extend beyond the plant itself, affecting the other organisms interacting with the plant [4]. Changes in plant physiology can affect the availability and quality of plant resources as food for herbivorous insects [5,6]. Fluctuations in the types and quantities of metabolites produced by plants under changing environmental conditions impact the feeding patterns, reproductive cycles, and overall population dynamics of herbivorous insects.

Elevated atmospheric CO_2_ concentrations not only act as a driver of climate change but also directly affect plant physiology and metabolism. Elevated CO_2_ concentrations impact the photosynthetic rate, water use efficiency, and nutrient concentrations in plants [7,8]. Previous studies have highlighted that elevated CO_2_ levels often promote plant growth through carbon fertilization, particularly in crop plants such as wheat, tomato, and soybeans [9,10,11]. They showed increased photosynthetic rates, enhanced biomass production, and altered allocation patterns of carbohydrates. These often result in increased leaf area, plant height, and overall biomass accumulation [12]. However, responses to increased CO_2_ levels were dissimilar among taxa [13]. In the case of *Aristolochia contorta* Bunge, belonging to Aristolochiaceae, growth reduction in elevated CO_2_ conditions has been reported [14]. *Aristolochia contorta* is an herbaceous perennial vine that mostly inhabits waterfront areas [15]. Height and photosynthetic rates of the species grown under elevated CO_2_ levels were significantly lower compared to those grown in ambient CO_2_ levels. Further investigation is required to better understand the responses of the plant to elevated CO_2_ levels.

The significant reduction in the growth of *A. contorta* in the elevated CO_2_ level suggested that other metabolic processes of *A. contorta*, including its growth and defense mechanism, could also be affected by increased CO_2_ concentration. *Aristolochia contorta* is known for producing aristolochic acid, a defensive secondary metabolite against herbivores [16,17]. Despite this, the *Sericinus montela* Gray (Papilionidae) specially target and feed on this plant [18]. The larvae of *S. montela* incorporate the aristolochic acid into their bodies, utilizing it as a defense mechanism against predators [19,20]. It is understood that changes in plant quality can influence the condition of specialist herbivores in response [4]. As such, variations in the metabolites of *A. contorta* caused by elevated CO_2_ level could affect the growth and survival of *S. montela*. *Sericinus montela* is classified as a vulnerable species in the red data book of Korea [21]. Considering the close ecological relationship between the plant and the butterfly, it is essential to predict how the dynamics of these two species will change in response to elevated CO_2_.

The response of *A. contorta* to herbivory, when grown under the elevated CO_2_ conditions, has been previously studied, particularly focusing on secondary metabolites and phytohormones [14]. Although the growth of *A. contorta* significantly declined in elevated CO_2_ levels, there was evidence of stronger defense response in that group when exposed to herbivory. Jasmonic acid, known for its role in activating defense responses, was significantly higher under the elevated CO_2_ than ambient CO_2_ group in first-year plants. Furthermore, the first-year senescent plants exhibit a 30-fold higher aristolochic acid I compared to the ambient CO_2_ group.

The inhibition of *A. contorta*’s growth under elevated CO_2_ conditions necessitates a thorough examination of the plant’s metabolic responses in such environments, with a specific focus on primary metabolites which directly influencing growth. Since primary metabolites are involved not only in growth but also in numerous other metabolic processes, these changes can affect other metabolic activities of the plant, such as the defense response [22,23]. Resistance and tolerance are two fundamental components of induced defense mechanisms of plants against herbivory [22,23,24,25]. Resistance involves the ability of plants to deter herbivores or reduce the damage through various means, such as the production of physical barriers or chemical deterrents. This strategy makes the plant less palatable or more difficult for the herbivore to consume [26]. Tolerance refers to a plant’s capacity to withstand or recover from the damage inflicted by herbivores. Instead of preventing herbivory, tolerance mechanisms allow plants to maintain or quickly regenerate their growth and reproductive output despite being consumed [27,28]. Plants simultaneously invest resources in both resistance and tolerance, thereby displaying a composite strategy of defense [29,30].

What defensive strategy does *A. contorta* adopt? As a perennial plant, *A. contorta* has the ability to store metabolites produced during the growing season to its roots for the following year. Moreover, the specific herbivory by *S. montela* is highly destructive to the above-ground parts of *A. contorta*. The larvae of *S. montela* hatch in clusters and consume significant portions of the leaves during their growth period [31]. Direct observations have shown that the leaves near the hatching sites of *S. montela* are nearly entirely eaten, particularly in small plants, leaving only the stems. Faced with such extensive consumption, *A. contorta* might lean towards a tolerance defense strategy, such as reallocating nutrients to its roots as a systemic response. On the other hand, research has shown that herbivory of *S. montela* triggers an increase in defensive secondary metabolites in the leaves of *A. contorta*, such as magnocurarine and magnoflorine [32,33], signifying a resistance defense response [34].

The investment in these defensive strategies varies depending on the environmental conditions of the plant. Since a rapid reduction in growth has been observed in elevated CO_2_ environments, the investment in defense strategies may also change under this condition. Because the defense strategies of *A. contorta* under elevated CO_2_ conditions have been studied in terms of secondary metabolites and phytohormone levels [14], investigating primary metabolites, which are directly involved in growth and also play a role in defense, is necessary. By analyzing the variations of primary metabolites under elevated CO_2_ and herbivory conditions, we can identify how the plant utilizes or stores substances involved in growth and defense.

We aim to investigate the primary metabolic responses of *A. contorta* under the following conditions: firstly, an elevated CO_2_ concentration as an abiotic factor; secondly, herbivory by *S. montela* as a biotic factor; and thirdly, the interaction of these two factors. Our focus will be on the responses of local and systemic inductions by tissue types, including local leaves directly subjected to simulated herbivory, systemic leaves not directly subjected to herbivory, and roots as another systemic tissue. This research will contribute to understanding how defense response of plants to specialist herbivore is affected by elevated CO_2_ conditions.

## 2. Results

### 2.1. Growth Characteristics and Chlorophyll Content of the Plants

When comparing the growth characteristics, plants in elevated CO_2_ conditions (eCO_2_; EC: Elevated CO_2_ and Control, EH: Elevated CO_2_ and Herbivory treatment) displayed a significant reduction in size and mass compared to those in ambient CO_2_ (aCO_2_; AC: Ambient CO_2_ and Control, AH: Ambient CO_2_ and Herbivory treatment). Plants in eCO_2_ were shorter, with an average height of 43.2 cm, compared to 72.8 cm in the aCO_2_ group (Figure 1a). Similarly, the dry weight and leaf area was lower in the elevated group (0.14 g, 28.8 cm^2^) than the ambient group (0.54 g, 108.5 cm^2^; Figure 1b,e). However, the lengths of the roots and the number of leaves were similar under two CO_2_ conditions (eCO_2_: 17.3 cm, 30.5 leaves; aCO_2_: 19.1 cm, 33.0 leaves; Figure 1c,d). Specific leaf area (SLA), measured as the ratio of leaf area to leaf dry mass, was lower in plants under elevated CO_2_ (eCO_2_: 36.2 m^2^∙kg^−1^, aCO_2_: 49.3 m^2^∙kg^−1^), indicating denser but smaller leaves (Appendix A). The chlorophyll content showed no significant difference across the four groups, although a notable interaction between CO_2_ levels and herbivory treatment was observed (Appendix A). Herbivory appeared to slightly reverse the reduction in chlorophyll content associated with elevated CO_2_ (23.68 SPAD in EC, 26.80 SPAD in AC, 25.53 SPAD in EH, 23.06 SPAD in AH; Figure 1f).

### 2.2. Carbon and Nitrogen Concentrations

The two-way analysis of variance of carbon and nitrogen concentrations revealed several significant effects of CO_2_ levels and the combined effect of CO_2_ with herbivory. Both CO_2_ levels and the combination of CO_2_ with herbivory significantly influenced the carbon concentrations (weight percent; wt%) in the systemic leaves (SL; 1st to 4th leaves; Figure 2). Particularly, systemic leaves of plants subjected to ambient CO_2_ with herbivory (AH) exhibited higher carbon concentrations compared to other groups. In contrast, carbon levels were lower in both the local leaves (LL; 5th to 6th leaves where herbivory was applied) and roots (R) of plants in elevated CO_2_ environments. Nitrogen levels in the roots were higher under elevated CO_2_ conditions. The carbon to nitrogen (C:N) ratio in the roots was lower in elevated CO_2_ conditions.

### 2.3. Primary Metabolite Concentrations

To assess the metabolic changes in *A. contorta*, gas chromatography-mass spectrometry (GC-MS) method was used to measure the types and quantities of primary metabolites. Samples were taken from three plant tissues: systemic leaves (SL), local leaves (LL) and roots (R), across the four experimental groups (AC, AH, EC, EH). A total of 46 metabolites were identified and categorized into 4 groups: sugars and sugar alcohols, amino acids, organic acids, and miscellaneous. There were 15 sugars and sugar alcohols, 10 amino acids, 11 organic acids, and 10 other metabolites (Appendix A). Principal component analysis (PCA) was conducted to examine the patterns of metabolite concentrations across different tissues. This analysis revealed distinct clusters, with roots clearly separating from leaf tissues (Figure 3a). Systemic and local leaves displayed similar metabolic profiles. The first principal component (PC1) accounted for 77.9% of the total variance, and the second component (PC2) explained another 13.0%. A one-way multivariate analysis of variance (MANOVA) confirmed significant differences in metabolite profiles based on tissue types, with principal components 1–45 as dependent variables (Table 1). The PCA results for primary metabolites of each tissue, based on the treatment groups, did not exhibit clear clustering (Figure 3b,c,d).

#### 2.3.1. Effects of Elevated CO_2_ on Primary Metabolites

Elevated CO_2_ levels significantly altered the concentrations of several metabolites across tissue types. In comparing the AC (Ambient CO_2_, Control without herbivory treatment) and EC (Elevated CO_2_, Control without herbivory treatment) groups, there were significant changes in twelve metabolites: nine in SL, two in LL, and five in R (Figure 4). In terms of sugars and sugar alcohols, the concentrations of glucose and lactose were reduced in the roots under elevated CO_2_ (Figure 4d,e), while myo-inositol increased in systemic leaves (Figure 4i). For amino acids, tryptophan levels rose in all tissue types under elevated CO_2_ conditions (Figure 4o). Among the organic acids, linoelaidic acid level was higher in systemic leaves in EC (Figure 4s), and there was a mixed response of monopalmitin, which was higher in systemic leaves but lower in roots in EC (Figure 4t). Palmitic acid showed a similar pattern: higher in SL, but lower in R in EC (Figure 4u). Additionally, glyceryl-glycoside and triacontanol concentrations were higher in both SL and LL in elevated CO_2_ (Figure 4x,ab). The concentrations of octacosanol and tocopherol were higher in SL in elevated CO_2_ (Figure 4y,aa).

#### 2.3.2. Effects of Herbivory on Primary Metabolites under Ambient CO_2_ Conditions

Herbivory treatment significantly altered the concentrations of five primary metabolites. When comparing plants treated with herbivory under ambient CO_2_ conditions (AH) to those without herbivory (AC), significant differences in the levels of two metabolites in SL, one in LL, and three in R were observed. For sugars and sugar alcohols, the concentration of pinitol was lower in AH than in AC in SL (Figure 4k). Although there were no significant differences between individual groups, the sucrose concentration in the roots tended to increase with herbivory treatment (Figure 4f).

In amino acids, glycine concentration was higher in AH than AC in LL (Figure 4n). Among organic acids, monopalmitin showed lower concentrations in both SL and R in AH (Figure 4t), while the concentrations of triacontanol in the roots were higher in AH than in AC (Figure 4ab). The concentration of phosphoric acid was significantly affected by herbivory treatment in all tissue types (Figure 4ae).

#### 2.3.3. Combined Effects of Elevated CO_2_ and Herbivory on Primary Metabolites

The interaction of elevated CO_2_ and herbivory treatments significantly affected the primary metabolites of *A. contorta*, particularly in systemic tissues. Comparing plants treated with both elevated CO_2_ and herbivory (EH) to those with elevated CO_2_ but no herbivory (EC), notable differences in 17 metabolites were observed: 15 in SL and 4 in R. In the category of sugars and sugar alcohols, concentration of galactinol, myo-inositol, and ribitol were lower in the EH group compared to EC group in SL (Figure 4h,i,j). For amino acids, levels of aspartic acid and glutamic acid decreased, while glycine increased in EH compared to EC in SL (Figure 4l,m,n). Among organic acids, butanedioic acid, monopalmitin, and palmitic acid showed decreased levels in EH in SL (Figure 4q,t,u), whereas lactic acid and stearic acid were higher in EH in R (Figure 4r,v). Additionally, concentrations of linoelaidic acid, sitosterol, and caffeic acid were lower in EH in SL (Figure 4s,z,ad). Glyceryl-glycoside was lower in EH in both SL and LL (Figure 4x). The phosphoric acid concentration was lower SL but higher R in EH compared to the EC group (Figure 4ae).

Overall, the number of metabolites with significant differences was highest in SL (26), followed by R (12), and LL (3). In terms of treatment, the combined effect of elevated CO_2_ and herbivory induced the most significant changes in metabolites, with a total of 19 significant differences observed. This contrasted with the lesser impact seen with herbivory alone, which showed the fewest differences (6). Additional data on primary metabolites not mentioned in the text can be found in Appendix A, where the original peak intensities were displayed, regardless of transformation to achieving normality. 

## 3. Discussion

### 3.1. Effects of Elevated CO_2_: Growth Inhibition and Changes of Stress-Related Primary Metabolites

The effects of elevated CO_2_ are estimated by comparing groups with different CO_2_ concentrations: one under ambient CO_2_ level (AC; Ambient CO_2_, Control without herbivory treatment) and another under elevated CO_2_ level (EC; Elevated CO_2_, Control without herbivory treatment). Typically, plants exhibit increased photosynthesis under elevated CO_2_ conditions, leading to higher carbon fixation rates and increased growth. However, *A. contorta* showed inhibited growth characteristics, contrasting with the general expectation. The height, dry weight, and total leaf area were all lower in the elevated CO_2_ group than the ambient CO_2_ group (Figure 1a,b,e). Furthermore, the carbon concentration within the plant was significantly lower in the EC group in LL and R compared to AC (Figure 2a). Previous study exposing *A. contorta* to the elevated CO_2_ concentration noted reduced maximum carboxylation rate (Vc_max_), indicating decreased rubisco activity [34]. Additionally, this study found that the amount of chlorophyll, known to correlate with the photosynthetic capacity, was slightly lower in the EC group than in the AC group (Figure 1f) [35,36]. This suggests that elevated CO_2_ level did not enhance *A. contorta*’s photosynthetic capacity and carbon fixation, potentially acting as a stressor instead. The diminished carbon fixation and growth could result from *A. contorta* diverting energy to produce non-essential metabolites for growth in response to elevated CO_2_ concentrations. Indeed, concentrations of several abiotic stress-response-related metabolites were significantly higher in the EC group than in the AC group. Myo-inositol, involved in the production of stress-related molecules [37], was significantly higher in EC than in AC in SL (Figure 4i). Octacosanol, associated with stress resistance by maintaining an outer protective layer, epicuticular waxes of leaves [38,39], was higher in EC in leaves (Figure 4y). Galactinol is a precursor for stress tolerance molecules such as raffinose family oligosaccharides (RFOs) which are involved in abiotic stress responses, including drought and cold [40]. The concentration of galactinol under elevated CO_2_ conditions had not been previously studied, but was found to be higher in EC in leaves (Figure 4h). The concentration of tocopherol, which is involved in stress tolerance by deactivating the reactive oxygen species (ROS) [41], was higher in EC than in AC in SL (Figure 4aa). Triacontanol, which modulates the activation of stress tolerance molecules, exhibited a higher concentration in EC in leaves (Figure 4ab) [42].

### 3.2. Effects of Herbivory: Local and Systemic Responses

When plants are attacked by insect herbivores, they often trigger a response that involves the production of defense metabolites [43]. Plants initiate signaling pathways to enhance defense mechanisms, thereby activating the production of specialized metabolites in response to herbivore attacks. This study used simulated herbivory methods to induce defense responses, but limitations exist as they do not fully mimic actual herbivore bites or account for ecological interactions, requiring careful interpretation of results [44]. The local leaves that are directly attacked may produce a higher concentration of defensive compounds at the site of damage [45,46]. This localized response usually aims to confine the damage and repel the herbivore [47]. In the local leaves (LL) of *A. contorta*, glycine exhibited a significantly higher concentration in AH (Ambient CO_2_, with Herbivory treatment) than in AC (Ambient CO_2_, Control without herbivory treatment; Figure 4n). The glycine-rich proteins (GRPs) superfamily participates in the mechanisms of signaling and cellular stress response [48]. Additionally, allose, known to upregulate defense-related and pathogenesis-related protein genes [49], was found at slightly higher concentrations in AH than in AC (Figure 4b). Triacontanol, involved in modulating stress-related signaling pathways [42], showed slightly higher concentrations in AH groups than in AC groups in LL (Figure 4ab).

*Aristolochia contorta* triggered a systemic response as well as localized response throughout the plant. When comparing the concentrations of primary metabolites in the group that received herbivory treatment and the group that did not, under ambient CO_2_ conditions (AH vs. AC), more significant differences in primary metabolites were observed within systemic tissues such as systemic leaves (SL; 2) and roots (R; 3) than in local leaves (LL; 1) where herbivory was directly applied. This indicates that the plant responded systemically to herbivory. The systemic response often involves signaling molecules such as jasmonic acid that travel from the damaged leaf to other parts of the plant [50,51]. In the previous study of *A. contorta*, the increase in jasmonic acid concentration was reported when herbivory of *S. montela* was treated [14]. The movement of jasmonic acid triggers a systemic defense response in other leaves, inducing the production of defense-related metabolites even in undamaged parts.

In systemic leaves (SL), the concentration of total soluble sugars, including allose, cellobiose, fructose, galactopyranose, glucose, lactose, maltose, sucrose, and trehalose, was slightly higher in AH than AC (Figure 4a). The signaling role of sugars in defense response by inducing resistance genes has been reported [52]. Trehalose, known to act as an elicitor of defense responses [53], exhibited slightly higher levels in AH than AC in SL (Figure 4g). Threonic acid, which increases in response to wounding stress [54], was slightly higher in AH than in AC in SL (Figure 4w). The concentrations of triacontanol, which is involved in regulating stress-related signaling pathways [42], was higher in AH than in AC in systemic leaves, similar to the patterns observed in local leaves (Figure 4ab). Tyrosine roles as a precursor of aristolochic acid, which is an important defensive second metabolite of *A. contorta* [17,55]. The concentration of tyrosine was slightly higher in AH compared to AC in SL (Figure 4p).

Recent studies have highlighted the critical role of roots in defense responses [56,57,58]. In the case of *A. contorta*, a perennial herb, examining responses of roots is essential for understanding the systemic responses. Signaling compounds triggered by herbivore attack can travel to the roots, prompting changes in root exudates and potentially influencing interactions with soil microbes or even signaling neighboring plants about potential threats [59]. In the roots (R) of *A. contorta*, several metabolites exhibited different concentrations between the AH and AC groups. Galactinol, associated with signaling during abiotic and biotic stress conditions [40], displayed slightly higher concentrations in AH than AC (Figure 4h). Elevated CO_2_ lowers the concentration of total soluble sugars in R (Figure 4a). The concentrations of triacontanol and allose were higher in AH than in AC in R, exhibiting a similar pattern to that observed in SL and LL (Figure 4b,ab).

### 3.3. Combined Effects of Elevated CO_2_ and Herbivory: Shifts in Defense Response

In our investigation of herbivory-induced responses under varying CO_2_ conditions, we observed differences in responses between the ambient CO_2_ (aCO_2_) and the elevated CO_2_ (eCO_2_) groups. The concentrations of primary metabolites exhibited significantly greater differences in the eCO_2_ group subjected to herbivory treatment compared to those in the aCO_2_ group. A comparison of groups grown under elevated CO_2_ conditions with and without herbivory treatment (EH vs. EC) revealed significant differences in 17 metabolites. Conversely, the comparison between the ambient CO_2_ groups with and without herbivory treatment (AH vs. AC) revealed only five metabolites with statistically significant differences. These results underscore the significant influence of elevated CO_2_ levels in restructuring plant responses to herbivory, leading to more pronounced shifts in primary metabolite concentrations under elevated CO_2_ conditions when exposed to herbivory.

When herbivory treatment was applied to both ambient CO_2_ and elevated CO_2_ groups, we observed different trends in several primary metabolites. Comparing AH with AC, the concentration of total soluble sugars was slightly higher in AH across all tissue types (Figure 4a). Conversely, when comparing EH with EC, the concentration was lower in EH in leaves but higher in roots. The concentrations of sucrose in the leaves exhibit similar responses, showing a slight increase with herbivory treatment in ambient CO_2_ groups but a decrease in elevated CO_2_ groups in leaves (Figure 4f). The concentration of glucose in roots (R) exhibited an opposite pattern. It was lower in AH than in AC, but higher in EH than in EC (Figure 4d). Several signaling-related metabolites under stress conditions exhibited varied trends in their herbivory-induced responses between ambient and elevated CO_2_ groups. Galactinol, a precursor of RFOs, functions as a signaling molecule in response to pathogen attacks and wounding [40]. Although there was no meaningful difference in galactinol concentration between AH and AC, the concentration was lower in EH than in EC in SL, but higher in R (Figure 4h). As mentioned, threonic acid is known to increase wounding stress [54]. It was slightly higher in all tissue types in AH than AC, but slightly lower in all tissue types in EH than EC (Figure 4w). Myo-inositol, involved in the production of stress-related molecules [37], did not exhibited a meaningful difference between the group AH and AC (Figure 4i). However, it exhibited lower concentration in leaves, particularly in SL in EH compared to EC. As indicated, the superfamily of glycine-rich proteins (GRPs) engaged in stress response mechanisms [48]. The concentration of glycine was significantly higher in AH than in AC in LL, but this difference was not observed in SL and R (Figure 4n). In EH, compared to EC, the concentration was higher in SL and R. Regarding the elicitor metabolites involved in defense response, trehalose was slightly higher in AH than in AC in leaves but slightly lower in EH than EC (Figure 4g) [53]. Allose, involved in upregulating pathogen-related genes [49], was marginally higher in AH compared to AC in SL but was slightly lower in EH compared to EC (Figure 4b). Tyrosine, a precursor of aristolochic acid [55,60], showed higher levels in AH than AC in SL but was comparable in SL and higher in R in EH than EC (Figure 4p). Dopamine, another precursor of aristolochic acid [60], showed no meaningful differences between the ambient CO_2_ groups in SL but was slightly lower in EH than EC in SL (Figure 4ac).

Numerous studies have addressed the reduced photosynthetic rate of plants in response to herbivory [61,62]. This reduction is often attributed to enhanced water loss and stomatal closure at sites of leaf attack. These studies reported changes in the expression of genes involved in photosynthesis, leading to a reduced carbon fixation rate [63,64,65]. In our study, a decrease in chlorophyll content was observed in *A. contorta* under ambient CO_2_ conditions when subjected to herbivory by *S. montela*. However, in elevated CO_2_ environments, plants subjected to herbivory displayed higher chlorophyll content compared to the group without herbivory [66]. By regulating their chlorophyll content, plants optimize the use of light energy. This adaptation enables them to continuously adjust to their environment. Although the exact photosynthetic rate was not measured in this study, an increase in chlorophyll content might indicate an effort to enhance the carbon fixation rate, possibly facilitating the synthesis of defense-related substances [67]. Indeed, comparing the EH and EC groups revealed more significant differences in the concentration of primary metabolites than the comparison between AH and AC groups. This observation is consistent with findings from previous research, which have noted increases in secondary metabolites and phytohormones in *A. contorta* subjected to herbivory under elevated CO_2_ conditions [34]. In response to herbivory in the elevated CO_2_ environment, *A. contorta* seems to exhibit greater resistance responses by producing more stress-related metabolites than in the ambient CO_2_ environment. This outcome aligns with prior research, where several crop genotypes that did not exhibit a decrease in photosynthesis rates in the face of herbivory demonstrated more resistant responses [68,69].

Several defense-related primary metabolites, including galactinol, threonic acid, myo-inositol, glycine, trehalose, and allose, exhibited higher concentrations in the roots and lower concentrations in the leaves when *A. contorta* was exposed to herbivory under elevated CO_2_ conditions. This indicates a more pronounced response in the roots under elevated CO_2_ compared to ambient CO_2_. Since the concentrations of carbon did not increase in the roots in EH, it is difficult to attribute the rise of primary metabolites in the roots to storage purpose (Figure 2a,b). A previous study dealing with the secondary metabolites of *A. contorta* also reported a significantly higher concentration of aristolochic acid, a well-known defensive metabolite, in the roots compared to the leaves [70]. This suggests that *A. contorta* may invest more defensive responses in its roots than in its leaves. Specifically, the increased concentration of primary metabolites in the roots under elevated CO_2_ conditions indicates that these defensive responses may be intensified at higher CO_2_ level. To understand the specific functions and interactions of primary metabolites, and to elucidate why the roots exhibited a stronger response rather than the leaves, where herbivory of *S. montela* occurred, further investigation is needed. Nonetheless, the observed trends in metabolite concentrations offer valuable insights into the complex dynamics of plant defense responses under varying CO_2_ conditions.

## 4. Materials and Methods

### 4.1. Plant Material and Growth Condition

Seeds of *A. contorta*, collected from Pyeongtaek-si and Gapyeong-gun in 2020, were stored at 4 °C, and sown in March 2021 in a soil mixture similar to their natural habitat (2:1 volumetric ratio of sand to loam) [71]. Sprouts were transplanted into pots and those over 7 cm tall were selected in July. These plants were placed in hexagonal open top chambers (OTCs) in a rooftop greenhouse with natural light exposure. The average height of the plants in each chamber were equalized. A shading net was installed on the ceiling of the greenhouse with about 50% relative light intensity (RLI), a suitable shade for *A. contorta* [72]. The variance in light intensity among the OTCs was negligible (RLI 34.5% to 44.4%, *p*-value 0.24). Plants were maintained in OTCs for about 45 days, watered 3 times a week for saturation, and received uniform amounts of fertilizer. Temperature and relative humidity values of each chamber, measured by sensors (HOBO Pro v2, Onset, Bourne, MA, USA), were similar (29.0 to 29.3 °C, 68.5 to 72.8%).

### 4.2. CO_2_ and Herbivory Treatment

Plants were divided into two groups with different CO_2_ levels: ambient (A, aCO_2_) and elevated CO_2_ (E, eCO_2_; Figure 5). The concentration of elevated CO_2_ was set to 540 ppm, based on the Representative Concentration Pathway (RCP) 4.5 scenario [73]. Levels of CO_2_ were regulated using a system equipped with a sensor-transmitter coupled with a CO_2_ controller (0–2000 ppm CO_2_, SH-MVG260, Soha-tech, Seoul, Republic of Korea), solenoid valve, and CO_2_ gas tank (99.99%, 40L). In the aCO_2_ group, a similar ventilation system was installed, as air is emitted evenly from all directions of the chamber. Three OTCs were used for each CO_2_ concentration, totaling six OTCs. The actual CO_2_ concentrations, measured by CO_2_ data logger (CDL 210, Wöhler, Bad Wünnenberg, Germany), were 429.2 ppm in the aCO_2_ group and 525.7 ppm in the eCO_2_ group.

To induce the defense response in *A. contorta*, herbivory was mimicked using the oral secretion of specialist herbivore *S. montela* applied on the leaves of the plants. Insect materials were initially collected at the egg stage from common gardens in Pyeongtaek-si, Gyeonggi-do, Republic of Korea (37°04′06″ N, 127°00′27″ E) and raised until needed. The herbivory simulation was conducted in August at noon, aligning with the feeding time *S. montela* hatchlings [14]. For this simulation, the fifth and sixth leaves of *A. contorta* were wounded with a pattern wheel, and 20 μL of *S. montela*’s oral secretion, diluted 20 times with deionized water, was applied. This approach, involving wounding and the application of the herbivore’s oral secretion, replicates effective defense induction method from previous research [74,75,76]. The oral secretion was obtained directly from third-instar larvae or older using pipette tips.

The experiment was structured into four groups to assess responses influenced by CO_2_ and herbivory. The groups included: AC (ambient CO_2_, Control), AH (ambient CO_2_, Herbivory treatment), EC (elevated CO_2_, Control), and EH (elevated CO_2_, Herbivory treatment). Each group consisted of twelve *A. contorta* plants evenly distributed across three OTCs, with each OTC containing four plants from each treatment group. Plant samples were collected five hours after the herbivory treatment at noon to analyze metabolic changes facilitated by photosynthesis. Three parts of the plants were collected for analysis: (1) the first to fourth leaves from the top, (2) the fifth to sixth leaves, and (3) the roots. The fifth to sixth leaves, where herbivory treatment was applied, were collected as local leaves (LL), while the first to fourth leaves were collected to assess the response of systemic leaves (SL). The root (R) was collected to examine the response in the underground parts, considering the significance of the underground part of perennial herb [77].

### 4.3. Measuring Chlorophyll Content and Biomass

Chlorophyll content in the leaves was measured non-destructively using a chlorophyll meter (SPAD-502plus, Konica Minolta, Tokyo, Japan) before collection. It is influenced by environmental factors such as nutrient availability and environmental stresses [78]. In the group treated with herbivory, the chlorophyll content was measured five hours after the treatment. The relative content of chlorophyll is calculated from the absorbance values to obtain Soil Plant Analysis Development (SPAD) value [79].

After harvesting, growth metrics including the height of the above-ground part of the plant, the length of the roots, the number of leaves, the total leaf area, and the dry weight were measured. The total leaf area per plant was measured with LI-3000C portable leaf area meter and LI-3050C transparent belt conveyors (LI-COR, Lincoln, NE, USA). The height of the above-ground part, the length of the root, and the number of leaves were measured in 12 plants from each treatment group (a total of 48 plants). Measurements of leaf area and dry weight were conducted exclusively for three plants from each ambient and elevated CO_2_ groups, to avoid the destruction which could impact primary metabolites. Specific leaf area (SLA) was calculated as the ratio of leaf area to leaf dry mass.

### 4.4. Carbon and Nitrogen Concentrations

Carbon and nitrogen concentration were measured by tissues (SL, LL, R) to assess the relative resource availability of the plants and to examine the impact of the stresses on the plants [80,81]. For the measurements of nitrogen and carbon content, twelve plants from each treatment group (AC, AH, EC, EH) were exclusively prepared, and four plants from each group were pooled into a single sample. The total carbon and nitrogen concentration were analyzed using an elemental analyzer (Flash EA 1112, Thermo Fisher Scientific, Waltham, MA, USA), at the National Instrumentation Center for Environmental Management (NICEM) at Seoul National University. The concentration was calculated based on the dry weight of each tissue.

### 4.5. Metabolic Profiling of Primary Metabolite

The concentration of primary metabolites in 12 plants from each experimental group, a total of 48 plants across 4 groups, was measured (Figure 5). Three parts of the plant were collected: Local leaves (LL), Systemic leaves (SL), and Roots (R). To prevent the change of metabolite after harvest, harvested tissues were immediately placed into the liquid nitrogen and stored at −80 °C. The Gas Chromatography-Mass Spectrometry (GC-MS) method was used to measure the concentration of primary metabolites in samples. Before analysis, the samples were lyophilized. For lyophilization, the samples were ground using pellet pestles and motors (Kimble chase, Vineland, NJ, USA) with liquid nitrogen. Ten milligrams of each sample were lyophilized. After lyophilization, the samples were prepared for GC-MS analysis. They were extracted with 1 ml of 100% methanol (HPLC grade, Sigma-Aldrich, St. Louis, MO, USA) and vortexed 1 min. After that, they were sonicated for 40 min and centrifuged with 110× *g* for 30 s. The supernatants of each sample were collected and filtered through 0.45 μm PTFE (polytetrafluoroethylene) syringe filters (Membrane solution, Plano, TX, USA). After filtration, 100 μL of each sample was transferred to amber GC vials (Agilent, Santa Clara, CA, USA) and dried with nitrogen gas for 5 min. To the dried samples, 30 μL of 20,000 μg/mL methoxyamine hydrochloride in pyridine (Sigma-Aldrich, USA) and 50 μl of BSTFA (*N*,*O*-bis (trimethylsilyl) trifluoroacetamide; Alfa Aesar, Ward Hill, MA, USA) containing 1% of trimethylchlorosilane (Sigma-Aldrich, USA) were added for oximation. For quantification, 10 μL of 300 μg/mL 2-chloronaphtalene in pyridine (Sigma-Aldrich, USA) was added as an internal standard (IS). The samples were evaporated in 65 °C for 1 h, and 100 μL of each sample was transferred to another amber GC vials contained insert. Prepared samples were stored at 4 °C before GC-MS analysis.

The concentrations of primary metabolites were assessed using a Shimadzu gas chromatography system (GC-MS QP2020, Shimadzu-Corp., Kyoto, Japan). This system was equipped with an MSD detector, an autosampler, a split/splitless injector, an injection module, and GCMSsolution software (https://www.shimadzu.co.kr/products/gas-chromatograph-mass-spectrometry/gc-ms-software/gcmssolution/index.html). To identify metabolites in the samples, the National Institute of Standards and Technology (NIST) mass spectral search library was employed, using a threshold of greater than 80% match quality for peak assignment. Subsequently, normalization was conducted by dividing the peak area of each compound by that of the internal standard (IS), thus enabling comparison of the relative abundance of selected metabolites across samples.

### 4.6. Statistical Analysis

Statistical methods were used to analyze plant height, length, dry weight, leaf count, and area. First, an F-test determined the homogeneity of variance among groups, and either Welch’s t-test or an independent samples t-test was followed depending on the results. For primary metabolites, chlorophyll content, and carbon to nitrogen ratio, significant differences among the four groups (AC, EC, AH, EH) were analyzed using a two-way analysis of variance (ANOVA), followed by Tukey’s honestly significant difference (HSD) test for detailed comparison. Normality of data was checked using the Shapiro–Wilk test. Concentrations of metabolites with non-normal distribution were transformed using logarithmic or exponential methods to meet normality requirements (Appendix A). Principal component analysis (PCA) was performed to evaluate the effect of tissue types and treatment on concentration of primary metabolites. Differences among tissue types were verified using a one-way multivariate analysis of variance (MANOVA), using principle components as the variables. Statistical analysis and illustration of the results were conducted using R version 4.2.1 [82] with the packages ‘ggpubr’ [83], ‘Momocs’ [84], and ‘ggplot2’ [85].

## 5. Conclusions

Our research provides insight into how *Aristolochia contorta* responds to elevated CO_2_ levels and herbivory, revealing complex interactions between plant growth, defense mechanisms, and environmental changes. Contrary to the general expectation of enhanced growth, the growth of *A. contorta* was inhibited under elevated CO_2_, suggesting unique responses of species to increased CO_2_ levels. The presence of herbivory by *Sericinus montela* also acts as a stress factor, triggering both local and systemic defense responses in the plant.

The study showed that elevated CO_2_ conditions significantly influenced the plant’s metabolic response to herbivory, indicating a shift towards defense. This included increased concentrations of defense-related metabolites, especially in the roots, suggesting a potential intensification of defensive responses in the underground parts of the plant under elevated CO_2_ levels.

Our findings underscore the importance of considering both abiotic and biotic factors in understanding plant responses to environmental changes. The adaptive strategies of *A. contorta*, balancing resistance and tolerance, suggest a complex response mechanism to elevated CO_2_ and herbivory pressures. This research contributes to our knowledge of plant response to environmental changes, emphasizing the need for further studies to comprehend the broader implications of these interactions on ecosystem dynamics.

## Figures and Tables

**Figure 1 plants-13-01456-f001:**
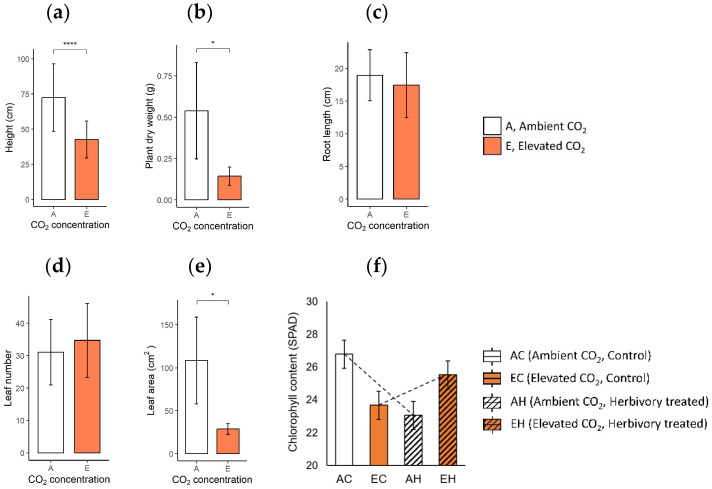
Growth characteristics of the plants in two CO_2_ concentrations are: (**a**) height, (**b**) dry weight, (**c**) root length, (**d**) leaf number, and (**e**) leaf area of the plant in Ambient CO_2_ (A) and Elevated CO_2_ (E) groups; (**f**) chlorophyll contents of four experimental groups across two CO_2_ levels and two herbivory treatments are: Ambient CO_2_ and Control (AC), Ambient CO_2_ and Herbivory treatment (AH), Elevated CO_2_ and Control (EC), and Elevated CO_2_ and Herbivory treatment (EH). The results of t-test are indicated on the graph in a, b, c, d, e, and the results of the two-way analysis of variance are presented through an interaction diagram on the graph f (* for *p*-value < 0.05; **** for *p*-value < 0.001).

**Figure 2 plants-13-01456-f002:**
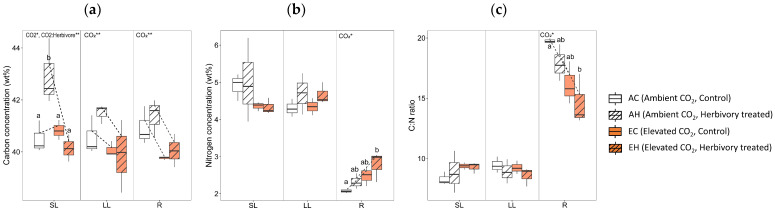
Carbon and nitrogen concentrations of three tissue types (local leaves that herbivory was treated, LL; systemic leaves, SL; roots, R) from four treatment groups (AC, AH, EC, EH). (**a**) carbon concentration, (**b**) nitrogen concentration, and (**c**) C:N ratio. The results of the two-way analysis of variance and Tukey’s HSD were presented through an interaction diagram, letters, and text above each graph (* for *p*-value < 0.05; ** for *p*-value < 0.01).

**Figure 3 plants-13-01456-f003:**
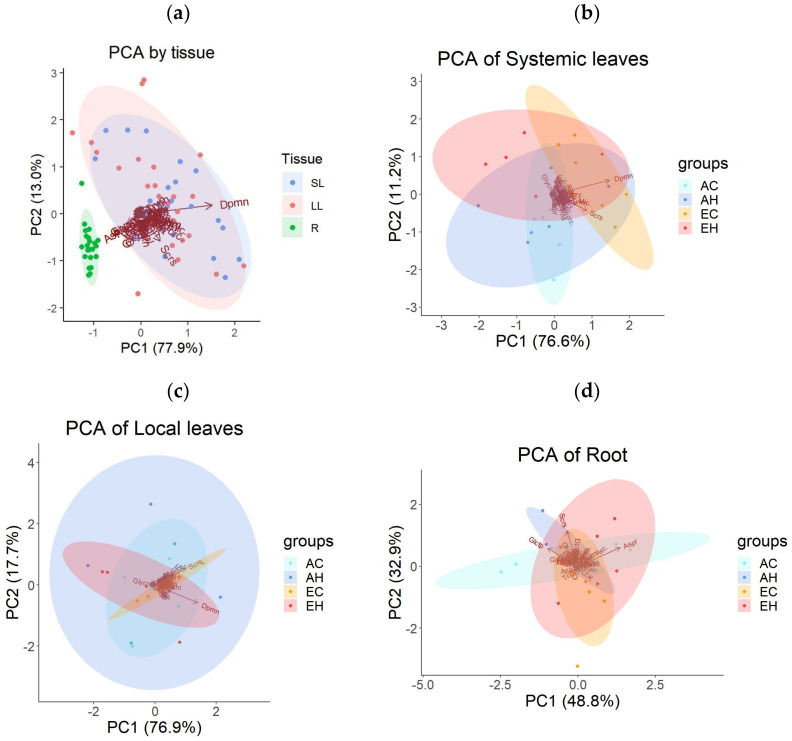
Results of the principal component analysis (PCA) on the concentrations of primary metabolites: (**a**) across three tissue types (local leaves treated with herbivory, LL; systemic leaves, SL; roots, R), (**b**) across four treatment groups (AC, AH, EC, EH) in SL, (**c**) in LL, and (**d**) in R. Abbreviations are used to describe the names of the primary metabolites.

**Figure 4 plants-13-01456-f004:**
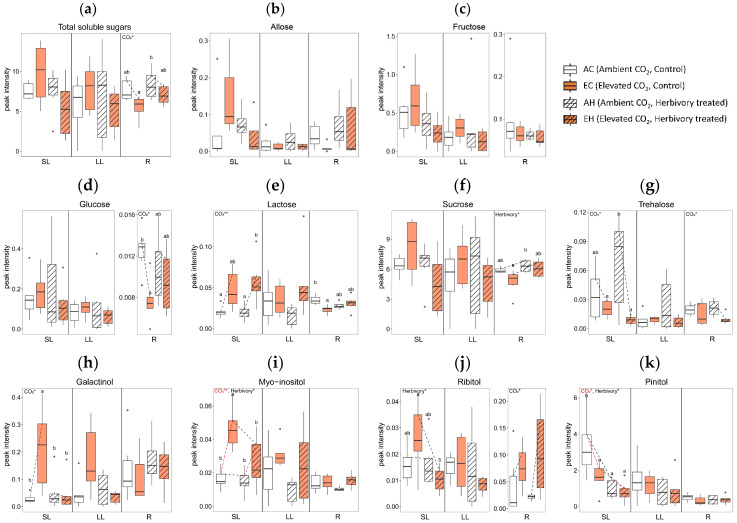
Concentration of primary metabolites by tissue types (local leaves that herbivory was treated, LL; systemic leaves, SL; roots, R) and four treatment groups (AC, EC, AH, EH). The results of the two-way analysis of variance and Tukey’s HSD are presented through an interaction diagram, letters, and text above each plot (* for *p*-value < 0.05; ** for *p*-value < 0.01).

**Figure 5 plants-13-01456-f005:**
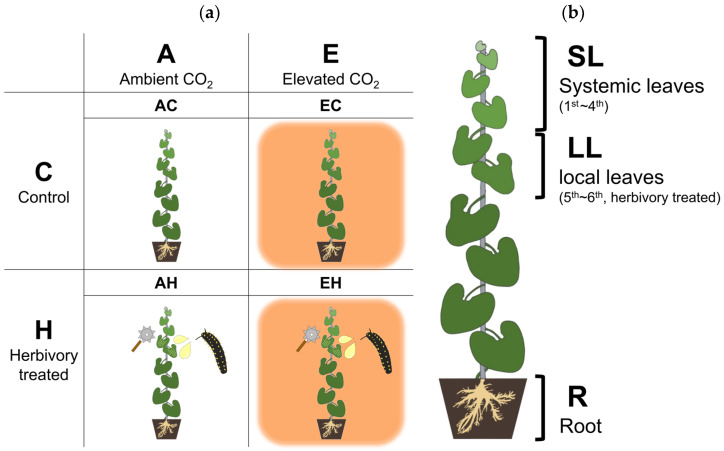
A schematic diagram of the experimental groups and collected tissues. (**a**) Four experimental groups, each with two CO_2_ levels (Ambient and Elevated CO_2_) and Herbivory treatments (Control and Herbivory-treated). (**b**) Three types of collected tissue. Systemic leaves (SL), Local leaves (LL), and Roots (R).

**Table 1 plants-13-01456-t001:** Results of the one-way multivariate analysis of variance (MANOVA) using principal components (PCs) of primary metabolite concentrations across tissue types as dependent variables. MANOVA retained seven eigenvalues.

	Df	Hotelling-Lawley	approx F	num Df	den Df	Pr (>F)
Tissue	2	10.74	99.726	14	260	<0.001
Residuals	137					

## Data Availability

The data are not publicly available due to the regulation of funding agency. The raw data supporting the conclusions of this article will be made available by the first author on request.

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
