# Peer review of "Primary Metabolic Response of *Aristolochia contorta* to Simulated Specialist Herbivory under Elevated CO_2_ Conditions"

_plants, 2024, doi:10.3390/plants13111456_

Round 1
Reviewer 1 Report
Comments and Suggestions for Authors
Overall, some changes are needed before this manuscript can be acceptable for publication. The authors need to better justify the rationale for their study, clarify their objectives, and explain their predictions for the responses they measured. The language about CO2 levels and climate change and photosynthetic rates needs to be corrected. The authors also need to correct or clarify some of their analyses.
The justification for measuring ‘primary metabolites’ isn’t very clear. How did the authors determine which metabolites to consider in this study? Why not target specific defense-related metabolites for these analyses? What were the predictions about how different classes of metabolites would be expected to change in response to the treatments?
The authors argue that elevated CO2 restricts plant growth which leads to an accumulation of excess sugars, which then signals plants to reduce photosynthesis, which then reduces growth. This argument appears to be quite circular and not well supported. First, it’s not clear why a plant would respond by reducing chlorophyll and whether this is an appropriate indicator of photosynthetic level in this case. Second, if a plant were to reduce photosynthesis, it would presumably then need to use the accumulated sugars to meet its growth requirements and would eventually begin photosynthesizing to fix additional carbon. The differences in chlorophyll level reported are small and it is not clear if any significant differences were detected (Figure 1f). If the authors wish to draw conclusions about photosynthetic rates, they should measure plant gas exchange.
Lines 17, 38 “Elevated levels of carbon dioxide” is not climate change, but is understood to be a driver of climate change and is associated with it. I think it’s important that the author’s make this differentiation. CO2 level in itself is not a characteristic of climate.
Lines 100-105 The authors should rephrase their objectives to be clearer. In Objectives 2 and 3, the authors can be more specific that their focus is on local induction in damaged leaves and systemic induction in non-damaged tissues.
In the results section, the plant growth characteristics are given for eCO2 and a CO2 and it is not clear if this is focusing only on the AC and EC treatments or whether this is pooling all plants (also those with herbivory) into the ambient and elevated groups. The authors should clarify this and use consistent treatment abbreviations.
It’s nice that the means are provided, but the authors should also include results from the statistical analyses here. For the leaf chlorophyll content, the authors should consider a two-way ANOVA (or similar non-parametric) to examine an interaction between the treatments, which the data suggest.
Figure 1f should include letters above bars to indicate differences
Figure 2, why does the PCA not include the CO2 treatments or the herbivory treatment? Did the authors pool all the treatments or only include one treatment here? It’s not clear.
Why would the leaf chlorophyll be expected to change within 5 hours? What is the prediction? Why was there an interaction?
Lines 222-224 Do the authors use a statistical analysis to compare SLA? I don’t see this in the results.
Lines 351-354 The authors did not measure photosynthetic rate or secondary metabolites so they cannot make these claims.
Line 417 specific herbivore should possibly be “specialist herbivore”. Otherwise they need to explain what is meant by “specific”.
Figure 4 This is a helpful diagram for understanding the experimental design.
A more specific timeline for the how long the CO2 treatments were applied and when the herbivory challenge treatment was applied would be helpful. The authors mention starting in July and ending in August, but it’s not clear how many weeks this experiment ran in total. How was the 5-hour post herbivore challenge time point selected? Is this significant in some way? Why not 1 hour or 1 week? The authors should explain their justification here.
Line 533, I don’t think it’s appropriate to say that you observed reduced photosynthesis here because of the lower chlorophyll content. The authors should measure actual gas exchange to draw this conclusion.
The conclusion section is overly wordy and can be summarized more concisely with the key points of the paper.
Comments on the Quality of English LanguageOverall fairly well written with only some minor English grammar issues, mostly in the discussion section.
Author Response
Comments
- Clarify their objectives, and explain their predictions for the responses they measured. The language about CO2levels and climate change and photosynthetic rates needs to be corrected.
- As you suggested, we revised introduction section. The focus of the objectives presented. We clarify the elevated CO2 and climate change.
Revised manuscript L127-134:
We aim to investigate the primary metabolic responses of A. contorta under the following conditions: first, an elevated CO2 concentration as an abiotic factor; second, herbivory by S. montela as a biotic factor; and third, the interaction of these two factors. Our focus will be on the responses of local and systemic inductions by tissue types, including local leaves directly subjected to simulated herbivory, systemic leaves not directly subjected to herbivory, and roots as another systemic tissue. This research will contribute to understanding how defense response of plants to specialist herbivore is affected by elevated CO2 conditions.
- The justification for measuring ‘primary metabolites’ isn’t very clear.
- As you pointed out, we acknowledge that we did not clearly present the rationale for analyzing primary metabolites. We enhance the rationale of measuring primary metabolites in the introduction section. We revised the introduction section in the context that 'In the situation of decreased growth seen in elevated CO2, we wanted to observe the metabolites related to growth, and in the situation where growth has noticeably decreased, we thought that this would also affect the defense response. Since previous studies have researched secondary metabolites and phytochromes under the same conditions, we conducted our research focusing on primary metabolites’.
Revised manuscript L70-126
- The authors argue that elevated CO2 restricts plant growth which leads to an accumulation of excess sugars, which then signals plants to reduce photosynthesis, which then reduces growth. This argument appears to be quite circular and not well supported.
- After considering the issues you pointed out, we concluded that the previously presented discussion was not reasonable. We deleted that section and presented a discussion in a different direction, additionally presenting the carbon and nitrogen concentration.
Revised manuscript L267-282:
Typically, plants exhibit increased photosynthesis under elevated CO2 conditions, leading to higher carbon fixation rates and increased growth. However, A. contorta showed inhibited growth characteristics, contrasting with the general expectation. The height, dry weight, and total leaf area were all lower in the elevated CO2 group than ambient CO2 group (Figure 1a,b,e). Furthermore, the carbon concentration within the plant was significantly lower in the EC group at LL and R compared to AC (Figure 2a). Previous study exposing A. contorta to the elevated CO2 concentration noted reduced maximum carboxylation rate (Vcmax), indicating decreased rubisco activity [32]. Additionally, this study found that the amount of chlorophyll, known to correlate with the photosynthetic capacity, was lower in the EC group than in the AC group (Figure 1f) [33,34]. This suggests that elevated CO2 level did not enhance A. contorta’s photosynthetic capacity and carbon fixation, potentially acting as a stressor instead. The diminished carbon fixation and growth could result from A. contorta diverting energy to produce non-essential metabolites for growth in response to elevated CO2 concentrations. Indeed, concentrations of several abiotic stress response-related metabolites were significantly higher in the EC group than in the AC group.
- Lines 17, 38 “Elevated levels of carbon dioxide” is not climate change, but is understood to be a driver of climate change and is associated with it. I think it’s important that the author’s make this differentiation. CO2 level in itself is not a characteristic of climate.
- As you suggested, we revised introduction section. The focus of the objectives presented. We clarify the elevated CO2 and climate change.
Revised manuscript L41-42:
One of the primary factors causing climate change is the increasing concentration of carbon dioxide (CO2).
Revised manuscript L55-56:
Elevated atmospheric CO2 concentrations not only act as a driver of climate change but also directly affect plant physiology and metabolism.
- Lines 100-105 The authors should rephrase their objectives to be clearer. In Objectives 2 and 3, the authors can be more specific that their focus is on local induction in damaged leaves and systemic induction in non-damaged tissues.
- As you suggested, we revised object section.
Revised manuscript L127-134:
We aim to investigate the primary metabolic responses of A. contorta under the following conditions: firstly, an elevated CO2 concentration as an abiotic factor; secondly, herbivory by S. montela as a biotic factor; and thirdly, the interaction of these two factors. Our focus will be on the responses of local and systemic inductions by tissue types, including local leaves directly subjected to simulated herbivory, systemic leaves not directly subjected to herbivory, and roots as another systemic tissue. This research will contribute to understanding how defense response of plants to specialist herbivore is affected by elevated CO2 conditions.
- In the results section, the plant growth characteristics are given for eCO2 and aCO2 and it is not clear if this is focusing only on the AC and EC treatments or whether this is pooling all plants (also those with herbivory) into the ambient and elevated groups. The authors should clarify this and use consistent treatment abbreviations.
- We acknowledge that our initial presentation may have been somewhat confusing. Height, length of the root, and leaf count were measured across all four treatment groups. Measurements of leaf area and dry weight were conducted exclusively for the ambient CO2 and elevated CO2 groups, as these assessments required the destruction of the plants, which could potentially impact primary metabolites. Therefore, the plants subjected to leaf area and dry weight measurement were not included from the primary metabolite analysis. This is the rationale for denoting these groups simply as ‘A’ and ‘E’ in our figure. We revised results and methodology section to clearly delineate the methods of measuring growth characteristics.
Revised manuscript L144-145:
The growth characteristics of the elevated CO2 group (eCO2: EC, EH) and the ambient CO2 group (aCO2: AC, AH) were compared.
Revised manuscript L508-515:
The height of the above-ground part, the length of the root, and the number of leaves were measured in twelve plants from each treatment group (a total of 48 plants). Measurements of leaf area and dry weight were conducted exclusively for three plants from each ambient CO2 and elevated CO2 groups, as these assessments required the destruction of the plants, which could potentially impact primary metabolites. Therefore, the plants subjected to leaf area and dry weight measurement were not included from the primary metabolite analysis. Specific leaf area (SLA) was calculated as the ratio of leaf area to leaf dry mass.
- the authors should also include results from the statistical analyses here. For the leaf chlorophyll content, the authors should consider a two-way ANOVA (or similar non-parametric) to examine an interaction between the treatments, which the data suggest.
- As you mentioned, a two-way ANOVA was performed. While each factor individually did not show significant results, there was a significant interaction effect between CO2 concentration and herbivory treatment. This indicates that the changes in chlorophyll content due to herbivory treatment vary depending on the CO2 Accordingly, these results have been added to the results section. The results table from the two-way ANOVA was added to Supplementary Table S2.
Revised manuscript L153-158:
Chlorophyll content was measured for the four groups (AC, AH, EC, EH). In herbivory treated groups, chlorophyll content was measured five hours after the herbivory treatment. The results of the two-way analysis of variance (ANOVA) indicated no significant differences among the four treatment groups (Supplementary Table S2); however, the interaction effect between CO2 concentration and herbivory treatment was found to be significant.
- Figure 1f should include letters above bars to indicate differences
- As you suggested, we added a letter to the graphs with significant differences (figure 1a,b,e).
- Figure 2, why does the PCA not include the CO2 treatments or the herbivory treatment? Did the authors pool all the treatments or only include one treatment here? It’s not clear.
- Previous Principal Component Analysis (PCA) that revealed the impacts of tissue types on the concentration of primary metabolite groups was conducted across all four treatment groups. We revised the methodology section to make it clear. The PCA results of treatment, were initially not presented due to their challenging interpretation. We included the results of PCA of treatment group in the Supplementary Figure S2.
Revised manuscript L569-572:
Principal component analysis (PCA) was performed to evaluate the influence of tissue types on the concentration of primary metabolites across four treatment groups. Additionally, PCA was also conducted to assess the effects of treatments on the primary metabolites within each tissue type.
- Lines 222-224 Do the authors use a statistical analysis to compare SLA? I don’t see this in the results.
- We added a value of specific leaf area to the Methods and Results section.
Revised manuscript L151-153:
Specific leaf area (SLA), measured as the ratio of leaf area to leaf dry mass, was on average 49.3 m2 ∙ kg-1 in the aCO2 group and 36.2 m2 ∙ kg-1 in the eCO2 group (Supplementary Figure S2).
Revised manuscript L515:
Specific leaf area (SLA) was calculated as the ratio of leaf area to leaf dry mass.
- Lines 351-354 The authors did not measure photosynthetic rate or secondary metabolites so they cannot make these claims.
- We removed that discussion and revised the direction of the discussion.
Revised manuscript L385-405
- Line 417 specific herbivore should possibly be “specialist herbivore”. Otherwise they need to explain what is meant by “specific”.
- We revised the miswording.
Revised manuscript L454-455:
To induce the defense response in A. contorta, an herbivory treatment imitating the specialist herbivore S. montela was conducted.
- A more specific timeline for the how long the CO2 treatments were applied and when the herbivory challenge treatment was applied would be helpful. The authors mention starting in July and ending in August, but it’s not clear how many weeks this experiment ran in total. How was the 5-hour post herbivore challenge time point selected? Is this significant in some way? Why not 1 hour or 1 week? The authors should explain their justification here.
- We aimed to apply herbivory treatment during the plant's active growth phase when it is most responsive to environmental cues. The seeds of contorta begins to sprout from April to May, achieving stable aboveground and root systems by around June. It then grows vigorously from July to August, before entering senescence in September, with the aboveground parts gradually detaching by around October. Considering the ontogeny of A. contorta, we planned to conduct experimental treatments and harvesting during its peak growth period in July and August.
Additionally, we aimed to apply herbivory treatment during the daytime, which is commonly known to be when the defense response is most recognizable and reactive.
Regarding the timeframe, previous studies have observed variations in the concentration of Jasmonic acid and several defensive metabolites within one hour after herbivory treatment in A. contorta. This study sought to explore the short-term metabolic shift at the resource allocation initiation stage. Given that herbivory was treated at noon, our aim was to examine the metabolic changes up to the point where sunlight could supply the energy and carbon for synthesis of defense-related metabolites via photosynthesis. Consequently, we chose to harvest five hours after the herbivory treatment.
We moved the sentence about how long the plants were grown to a more visible section and added details and rationale of the timing of herbivory treatment.
Revised manuscript L442-444:
The plants were cultivated in OTCs for approximately 45 days before harvest to conduct treatment during the active growth phase of A. contorta.
Revised manuscript L483-485:
Plants samples were collected five hours after the herbivory treatment at noon to examine the metabolic changes up to the point where sunlight could supply the energy and carbon necessary for the synthesis of defense-related metabolites via photosynthesis.
- Why would the leaf chlorophyll be expected to change within 5 hours? What is the prediction? Why was there an interaction?
- The timing for measuring chloroplasts was aligned with the timing for measuring primary metabolites. We added a rationale for the timeline of measuring primary metabolites to the Materials and Methods section.
Revised manuscript L483-485:
Plants samples were collected five hours after the herbivory treatment at noon to examine the metabolic changes up to the point where sunlight could supply the energy and carbon necessary for the synthesis of defense-related metabolites via photosynthesis.
- Line 533, I don’t think it’s appropriate to say that you observed reduced photosynthesis here because of the lower chlorophyll content. The authors should measure actual gas exchange to draw this conclusion.
- Accepting the points you raised, we revised the discussion and conclusion parts. We removed the part you pointed out from the conclusion section.
- The conclusion section is overly wordy and can be summarized more concisely with the key points of the paper.
- As you suggested, I simplified the conclusion part.
Revised manuscript L578-595
Reviewer 2 Report
Comments and Suggestions for Authors
The topic is quite interesting, and under-researched. However, the methods need to be better described. It seems that an unreplicated experimental design was used, which would not be adequate to properly test the hypothesis.
I did not find where the total number of chambers was given, although it seems to be just 4, with each treatment in only one chamber. If true, that is an unreplicated design, and would not be statistically valid. Hopefully I am mistaken, but it needs to be more clear.
On line 414, it seems like n = 5, which seems to contradict line 112, which gives 12 plants per treatment (although apparently all 12 in one chamber). Were the 4 OTC all in the same greenhouse compartment?
The huge decrease in dry mass and leaf area per plant caused by such a small increase in CO2 seems highly unlikely. Is there anything comparable in the literature? Any possible explanation? Perhaps some contamination in the CO2?
Author Response
Comments
- I did not find where the total number of chambers was given, although it seems to be just 4, with each treatment in only one chamber. If true, that is an unreplicated design, and would not be statistically valid. Hopefully I am mistaken, but it needs to be more clear.
On line 414, it seems like n = 5, which seems to contradict line 112, which gives 12 plants per treatment (although apparently all 12 in one chamber). Were the 4 OTC all in the same greenhouse compartment?
- We recognize that our previous description of the number of Open Top Chambers (OTCs) and plants of treatment groups was unclear. In the experimental design, a total of six OTCs was used: three assigned to the ambient CO2 (aCO2) group and three to the elevated CO2 (eCO2) group. Each of the four treatment groups—ambient CO2 control (AC), elevated CO2 control (EC), ambient CO2 with herbivory (AH), and elevated CO2 with herbivory (EH)—included twelve plants, distributed equally across the OTCs. Consequently, each OTC contributed four plants to each treatment group. We added this detail to the manuscript.
Revised manuscript L429-431:
The selected plants were placed into six hexagonal open top chambers (OTCs) with a height of 1.1 m and diameter of 1.3 m in the greenhouse.
Revised manuscript L480-482:
Each group included twelve A. contorta plants, distributed equally across the OTCs. Since three OTCs were utilized for each two CO2 concentration, each OTC contributed four plants to each treatment group.
- The huge decrease in dry mass and leaf area per plant caused by such a small increase in CO2 seems highly unlikely. Is there anything comparable in the literature? Any possible explanation? Perhaps some contamination in the CO2?
- Previous studies that grew contorta in elevated CO2 environments have already observed significantly reduced growth [Park, H. J., Nam, B. E., Lee, G., Kim, S. G., Joo, Y., & Kim, J. G. (2022). Ontogeny-dependent effects of elevated CO2 and watering frequency on interaction between Aristolochia contorta and its herbivores. Science of The Total Environment, 838, 156065.].

Reviewer 3 Report
Comments and Suggestions for Authors
The article “Primary metabolic response of Aristolochia contorta to specialist herbivore under elevated CO2 conditions” (plants-2938216) studies the effect of CO2 levels on Aristolochia contorta metabolites and its response to a specialist herbivore. The manuscript is well-written and the research fits the journal, but I have some concerns about the way the herbivory experiments were conducted and about not including plant secondary metabolites in the analysis; this methodology limits the application and interest of the results. Further explanations are needed on the choice of the methodology to mimic herbivory and on not including plant secondary metabolites in the analysis. If the authors have additional data on the plant secondary metabolites, it should be included in the manuscript.
The title is misleading and should be changed because no actual herbivores (S. montela) where used in the experiments. Instead authors attempted to simulate the effect of natural herbivory by mechanically wounding leaves with a pattern wheel and by applying a certain amount of oral secretion of S. montela diluted with deionized water, was applied.
Abstract
Line 20: Add the full name of Aristolochia contorta, i.e., Aristolochia contorta Bunge (Aristolochiaceae). The same for Sericinus montela.
Introduction
Given the importance of plant secondary metabolites in plant defense to herbivore, why were only primary metabolites included in the investigation?
Discussion
The authors should discuss the differences on the effect on plants between using an actual herbivore versus using mechanical wounding combined with oral secretion collected from the herbivore.
Material and Methods
Lines 422-424: Why were no actual S. montela larvae used in the experiment? How was the oral secretion of S. montela collected and why did the authors use these amounts?
Line 432: The authors should also explain why plants samples were collected five hours after treatment. Five hours is a relatively short time compared to the feeding that occurs in natural conditions of herbivory. Several studies have reported on the differences on the effect of herbivory on primary and secondary metabolites over time.
Author Response
Comments
- The manuscript is well-written and the research fits the journal, but I have some concerns about the way the herbivory experiments were conducted and about not including plant secondary metabolites in the analysis; this methodology limits the application and interest of the results.
- As you pointed out, we acknowledge that we did not clearly present the rationale for analyzing primary metabolites. We enhance the rationale of measuring primary metabolites in the introduction section. We revised the introduction section in the context that 'In the situation of decreased growth seen in elevated CO2, we wanted to observe the metabolites related to growth, and in the situation where growth has noticeably decreased, we thought that this would also affect the defense response. Since previous studies have researched secondary metabolites and phytochromes under the same conditions, we conducted our research focusing on primary metabolites’.
Revised manuscript L70-126
- The title is misleading and should be changed because no actual herbivores ( montela) where used in the experiments. Instead authors attempted to simulate the effect of natural herbivory by mechanically wounding leaves with a pattern wheel and by applying a certain amount of oral secretion of S. montela diluted with deionized water, was applied.
- As you pointed out, we revised the title as ‘simulated specialist herbivory’.
Revised manuscript L2-3:
Primary metabolic response of Aristolochia contorta to simulated specialist herbivory under elevated CO2 conditions
- Abstract
Line 20: Add the full name of Aristolochia contorta, i.e., Aristolochia contorta Bunge (Aristolochiaceae). The same for Sericinus montela. - We revised the text.
Revised manuscript L18-20:
We focused on Aristolochia contorta Bunge (Aristolochiaceae), a wild plant that exhibit growth reduction under elevated CO2 in the previous study. The plant has Sericinus montela Gray (Papilionidae) as a specialist herbivore.
- Introduction
Given the importance of plant secondary metabolites in plant defense to herbivore, why were only primary metabolites included in the investigation? - We revised the introduction to emphasize the rationale for measuring primary metabolites, and to note that secondary metabolites have been studied under the same condition.
Revised manuscript L70-126
- Discussion
The authors should discuss the differences on the effect on plants between using an actual herbivore versus using mechanical wounding combined with oral secretion collected from the herbivore.
Material and Methods
Lines 422-424: Why were no actual S. montela larvae used in the experiment? How was the oral secretion of S. montela collected and why did the authors use these amounts?
- Initially, obtaining a sufficient number of montela larvae for the herbivory treatment was challenging. Sericinus montela, a wild species classified as vulnerable in Korea, was difficult to acquire in large quantities. Moreover, given the life cycle of the butterfly, collecting eggs and rearing them to the larval stage for the herbivory treatment introduced complexities in timing and in preparing larvae of a controlled size and instar for the experiment. Consequently, we resorted to a well-established simulation method to effectively induce plant defense responses, which entails wounding and applying a specialist herbivore's oral secretion.
The oversight regarding the documentation of specific details and reference literature for the wounding and oral secretion treatment is acknowledged. These details have been added to the revised manuscript.
Revised manuscript L465-473:
To simulate natural herbivory, the fifth and sixth leaves of A. contorta were wounded with a pattern wheel, and 20 μl of the oral secretion of S. montela, diluted 20-fold with deionized water, was applied. This method of simulating herbivory, through wounding and applying the specialist herbivore's oral secretion, is recognized for effectively inducing plant defense responses [70]. The treatment volume replicates the defense induction used in prior studies, induced a plant defense response without the need for substantial leaf tissue removal [71,72]. The oral secretion of S. montela was directly collected from larvae of third-instar or older using pipette tips.
- Line 432: The authors should also explain why plants samples were collected five hours after treatment. Five hours is a relatively short time compared to the feeding that occurs in natural conditions of herbivory. Several studies have reported on the differences on the effect of herbivory on primary and secondary metabolites over time.
- We acknowledge the oversight and have revised the manuscript to include a detailed explanation of the timing of the herbivory treatment. Regarding the timeframe, previous studies have observed variations in the concentration of Jasmonic acid and several defensive metabolites within one hour after herbivory treatment in contorta. This study sought to explore the short-term metabolic shift at the resource allocation initiation stage. Given that herbivory was treated at noon, our aim was to examine the metabolic changes up to the point where sunlight could supply the energy and carbon for synthesis of defense-related metabolites via photosynthesis. Consequently, we chose to harvest five hours after the herbivory treatment.
Revised manuscript L483-485:
Plants samples were collected five hours after the herbivory treatment at noon to examine the metabolic changes up to the point where sunlight could supply the energy and carbon necessary for the synthesis of defense-related metabolites via photosynthesis.

Reviewer 4 Report
Comments and Suggestions for Authors
Majors:
- authors have treated plants by wounding plus applying oral secretion and used untreated plants as control. Another control by wounding but without application of secrete would have been nice
- light conditions in the greenhouse must be added
- biochemists should know that centrifugation values must be presented in g and not in rpm
- species names in the references must be checked for correct spelling and in italics
Minors:
- line 143: Supplementary Table S1
- line 443: subtitle in lowercase letters (as all others)
- line 464: metabolites
- line 467: were lyophilized
- lines 471 and 475: names of chemicals in lowercase letters (as all others)
Comments on the Quality of English Language
English is fine and the manuscript needs only few language and spelling corrections (see above).
Author Response
Comments
- Authors have treated plants by wounding plus applying oral secretion and used untreated plants as control. Another control by wounding but without application of secrete would have been nice.
- We agree with the point you raised. Initially, our plan included a control group treated with only diluted water during the research planning stage. However, due to budgetary constraints, this group was not analyzed. In future research, we will ensure that a better-designed experiment is conducted.
- Light conditions in the greenhouse must be added.
- As you pointed out, we added an explanation about light intensity.
Revised manuscript L432-438:
The greenhouse, located on the roof of the building, is managed under unshaded natural light. A shading net was installed on the ceiling of the greenhouse with about 50% relative right intensity, a suitable shade for A. contorta [71]. Light intensity measurements within the OTCs were conducted at two-hour intervals from 10:00 to 14:00. The variance in light intensity among the OTCs was negligible, as indicated by a p-value of 0.24 in the results of one-way analysis of variance; the actual relative light intensity values ranged from 34.5% to 44.4% at 14:00.
- Biochemists should know that centrifugation values must be presented in g and not in rpm.
- We revised the unit.
Revised manuscript L535-536:
After that, they were sonicated for 40 min and centrifuged with 110 g for 30 sec.
- Species names in the references must be checked for correct spelling and in italics.
- We revised the errors you pointed out, but some species names were not modified because they were incorrectly stated in the title of the original research, such as Sericinus montelus.
- line 143: Supplementary Table S1
- We revised the error you pointed out.
Revised manuscript L194-195:
There were 15 sugars and sugar alcohols, ten amino acids, eleven organic acids, and ten other metabolites (Supplementary Table S1).
- line 443: subtitle in lowercase letters (as all others)
- We revised it.
Revised manuscript L497:
4.3. Measuring chlorophyll content and biomass
- line 464: metabolites
- We modified the word to plural.
Revised manuscript L529-530:
Gas chromatography-mass spectrometry (GC-MS) method was used to measure the concentration of primary metabolites in samples.
- line 467: were lyophilized
- We modified the word to plural.
Revised manuscript L532-533:
Ten milligrams of each sample were lyophilized.
- lines 471 and 475: names of chemicals in lowercase letters (as all others)
- We modified the words to lowercase.
Revised manuscript L536-537:
The supernatants of each sample were collected and filtered through 0.45 μm PTFE (polytetrafluoroethylene) syringe filters (Membrane solution, Plano, USA).
Revised manuscript L539-542:
To the dried samples, 30 μL of 20,000 μg/ml methoxyamine hydrochloride in pyridine (Sigma-Aldrich, USA) and 50 μL of BSTFA (N,O-bis (trimethylsilyl) trifluoroacetamide; Alfa Aesar, Ward Hill, MA, USA) containing 1% of trimethylchlorosilane (Sigma-Aldrich, USA) were added for oximation.

Round 2
Reviewer 3 Report
Comments and Suggestions for Authors
The manuscript has been improved. However, as I mentioned earlier, the authors should discuss in the Discussion section the differences on the effect on plants between using an actual herbivore versus using mechanical wounding combined with oral secretion collected from the herbivore.
Also correct reference 34, line 686, the author's family name should be Park.
Author Response
- The authors should discuss in the Discussion section the differences on the effect on plants between using an actual herbivore versus using mechanical wounding combined with oral secretion collected from the herbivore.
- As you suggested, we mentioned the differences of the actual herbivory and simulated herbivory on the discussion section.
Revised manuscript L:
This study used simulated herbivory methods to induce defense responses, but limitations exist as they do not fully mimic actual herbivore bites or account for ecological interactions, requiring careful interpretation of results.
- Correct reference 34, line 686, the author's family name should be Park.
- As you pointed out, we revised the typing mistake.
Revised manuscript L:
34. Park, H.J. Effects of environmental factors on host plant and its specialist herbivore, Aristolochia contorta and Sericinus montela. Seoul National University Graduate School, 2021.
